# Enhancing Early GI Disease Detection with Spectral Visualization and Deep Learning

**DOI:** 10.3390/bioengineering12080828

**Published:** 2025-07-30

**Authors:** Tsung-Jung Tsai, Kun-Hua Lee, Chu-Kuang Chou, Riya Karmakar, Arvind Mukundan, Tsung-Hsien Chen, Devansh Gupta, Gargi Ghosh, Tao-Yuan Liu, Hsiang-Chen Wang

**Affiliations:** 1Division of Gastroenterology and Hepatology, Department of Internal Medicine, Ditmanson Medical Foundation Chia-Yi Christian Hospital, Chia-Yi 60002, Taiwan; 02601@cych.org.tw (T.-J.T.); vacinu@gmail.com (C.-K.C.); 2Department of Trauma, Changhua Christian Hospital, Changhua, No. 135, Nanxiao St., Changhua City, Changhua County 50006, Taiwan; 88847@cch.org.tw; 3Obesity Center, Ditmanson Medical Foundation Chia-Yi Christian Hospital, Chia-Yi 60002, Taiwan; 4Department of Mechanical Engineering, National Chung Cheng University, 168, University Rd., Min Hsiung, Chia-Yi 62102, Taiwan; karmakarriya345@gmail.com (R.K.); arvindmukund96@gmail.com (A.M.); 5Department of Biomedical Imaging, Chennai Institute of Technology, Sarathy Nagar, Chennai 600069, Tamil Nadu, India; 6Department of Internal Medicine, Ditmanson Medical Foundation Chia-Yi Christian Hospital, Chia-Yi 60002, Taiwan; cych13794@gmail.com; 7Computer Science and Engineering Department, Thapar Institute of Engineering & Technology, Patiala 147001, Punjab, India; dgupta1_be21@thapar.edu; 8Apex Institute of Technology, Chandigarh University, Ludhiana 140413, Punjab, India; ghosh.gargi001@gmail.com; 9Department of Pediatrics, Kaohsiung Armed Forces General Hospital, 2, Zhongzheng 1st. Rd., Lingya District, Kaohsiung City 80284, Taiwan; 10Department of Medicine, National Defense Medical University, No. 161, Sec. 6, Minquan E. Rd., Neihu District, Taipei City 11490, Taiwan; 11Department of Medical Research, Dalin Tzu Chi Hospital, Buddhist Tzu Chi Medical Foundation, No. 2, Minsheng Road, Dalin, Chia-yi 62247, Taiwan; 12Department of Technology Development, Hitspectra Intelligent Technology Co., Ltd., Kaohsiung 80661, Taiwan

**Keywords:** color calibration, deep learning, early diagnosis, endoscopy, gastrointestinal diseases, hyperspectral imaging, image enhancement, narrow-band imaging, spectral reconstruction, spectrum aided vision enhancer, while-light Imaging

## Abstract

Timely and accurate diagnosis of gastrointestinal diseases (GIDs) remains a critical bottleneck in clinical endoscopy, particularly due to the limited contrast and sensitivity of conventional white light imaging (WLI) in detecting early-stage mucosal abnormalities. To overcome this, this research presents Spectrum Aided Vision Enhancer (SAVE), an innovative, software-driven framework that transforms standard WLI into high-fidelity hyperspectral imaging (HSI) and simulated narrow-band imaging (NBI) without any hardware modification. SAVE leverages advanced spectral reconstruction techniques, including Macbeth Color Checker-based calibration, principal component analysis (PCA), and multivariate polynomial regression, achieving a root mean square error (RMSE) of 0.056 and structural similarity index (SSIM) exceeding 90%. Trained and validated on the Kvasir v2 dataset (*n* = 6490) using deep learning models like ResNet-50, ResNet-101, EfficientNet-B2, both EfficientNet-B5 and EfficientNetV2-B0 were used to assess diagnostic performance across six key GI conditions. Results demonstrated that SAVE enhanced imagery and consistently outperformed raw WLI across precision, recall, and F1-score metrics, with EfficientNet-B2 and EfficientNetV2-B0 achieving the highest classification accuracy. Notably, this performance gain was achieved without the need for specialized imaging hardware. These findings highlight SAVE as a transformative solution for augmenting GI diagnostics, with the potential to significantly improve early detection, streamline clinical workflows, and broaden access to advanced imaging especially in resource constrained settings.

## 1. Introduction

GID is a substantial global health concern, contributing significantly to morbidity and mortality. The World Health Organization (WHO) estimates that digestive diseases rank among the main causes of disability-adjusted life years globally and result in millions of deaths yearly [1,2]. Improving treatment outcomes, slowing down disease development, and increasing patient survival all depend on early, precise identification of GID [3,4]. However, minor mucosal and vascular alterations are typically difficult to detect with traditional imaging techniques, highlighting the need for enhanced diagnostic tools. Normal structures (pylorus, Z-line, and cecum), esophagitis, polyps, ulcerative colitis, dyed-lifted polyps, and dyed-resection margins are the six medically significant types of GI disorders that this study focuses on [5,6,7,8,9,10,11,12]. Every class marks a different diagnostic constraint. The pylorus regulates the passage of food from the stomach into the small intestine; the Z-line marks a critical transition point in the oesophagus where changes associated with Barrett’s oesophagus can occur; and a complete evaluation of the colon during endoscopy requires successful visualization of the cecum. Esophagitis, which is characterized by oesophageal inflammation, can proceed to Barrett’s oesophagus and, eventually, cancer if left untreated [13,14]. Ulcerative colitis is a chronic inflammatory illness raising the risk of colonic cancer; polyps are precancerous mucosal bulges with a risk of malignant transformation. In endoscopic mucosal resections, when precise visibility is required to guarantee full lesion removal and avoid recurrence, dyed-raised polyps and dyed-resection margins indicate treated areas [15]. By allowing for a comprehensive and non-invasive view of the GI tract, especially the small intestine, capsule endoscopy (CE) has revolutionized GI diagnosis [16,17]. Naturally traversing the digestive system, the capsule, a small wireless camera, captures hundreds of photos without the need for sedation or intubation. High patient tolerance, broad visual reach, and useful application in the diagnosis of small intestinal tumors, Crohn’s disease, and unexplained GI bleeding define its benefits [18,19]. But capsule endoscopy has drawbacks as well: lack of real-time control over capsule movement, inability to take biopsies, reliance on battery life, and most importantly, reliance on conventional WLI [20]. Particularly for identifying early or mild mucosal alterations, these restrictions lower its diagnostic sensitivity. One cannot emphasize the need for early identification in GID. Many GI diseases provide a critical window for intervention since they follow a slow, silent development from benign to malignant stages. For example, identifying and removing adenomatous polyps will completely stop colorectal cancer from starting. Early esophagitis management also helps to avoid problems such as Barrett’s oesophagus and related cancer. Early therapy can help to reduce inflammation in ulcerative colitis, hence lowering dysplasia and cancer risk. Therefore, technologies that can improve early-stage lesion detection offer significant therapeutic benefits, not only improved patient outcomes but also lower healthcare expenses related to late-stage disease management.

HSI produces comprehensive spatial and spectral data for every pixel by capturing photos over hundreds or thousands of spectral bands [21]. Spectral range guides HSI systems into visible (400–700 nm), near-infrared (700–1000 nm), and short-wave infrared (1000–2500 nm) imaging [22,23]. Because HSI may non-invasively detect biochemical and morphological changes, it has shown efficiency in biomedical applications including cancer detection, surgical guidance, and tissue viability assessment. HSI can perhaps show aberrant tissue patterns unseen under standard WLI in GI endoscopy. For ordinary clinical endoscopy, however, conventional HSI systems are frequently costly, heavy, and sluggish, making them unworkable. NBI uses the absorption properties of haemoglobin to improve mucosal view-through [24,25], whereas light at 540 nm (green) probes deeper tissues to view submucosal arteries, and light at 415 nm (blue) emphasizes superficial capillaries. Haemoglobin has large absorption peaks in the 450–510 nm region; hence, alternative wavelengths such as 450 nm and 510 nm have been investigated to further improve this contrast [26,27]. By darkening blood vessels relative to surrounding tissues, NBI greatly increases lesion visibility and helps detect early neoplastic alterations. For early-stage GI malignancies and inflammatory alterations, NBI routinely shows greater sensitivity, specificity, and general diagnostic accuracy than WLI. Notwithstanding its benefits, NBI is usually limited to specialized high-end endoscopic systems and cannot be used retroactively to previously obtained WLI pictures. Although both HSI and NBI have great advantages, taken separately each has certain drawbacks. While NBI systems are hardware-dependent and not found on many ordinary endoscopes, hyperspectral devices are expensive and complicated. SAVE was motivated to combine the merits of both technologies, leveraging the HSI comprehensive spectral data and the NBI vascular enhancement without the related hardware limits. From normal WLI images, SAVE computationally reconstructs hyperspectral information and synthetically generates narrow-band images targeted on diagnostically appropriate wavelengths [28]. On conventional and capsule endoscopy systems, this software-based method allows for improved imaging capabilities without requiring expensive equipment modifications. The SAVE enhances endoscopic visualization by combining the extensive spectral data from hyperspectral imaging with the focused contrast improvement of narrow-band imaging [29]. It allows doctors to focus especially on mucosal and vascular traits that are important for early illness diagnosis. SAVE also helps to perform retroactive analysis of current WLI data, thereby revealing possibly missed lesions and supporting longitudinal patient monitoring.

In this work, the SAVE performance was thoroughly validated using deep learning models trained on the Kvasir v2 dataset. We assessed classification performance among architectures including ResNet-50, ResNet-101, EfficientNet-B2, EfficientNet-B5, and EfficientNetV2-B0. SAVE-enhanced versus conventional WLI images were fully evaluated using key performance metrics including precision, recall, F1-score, SSIM, entropy, and peak signal-to-noise ratio (PSNR). Comparative results show that SAVE routinely increases classification accuracy across all studied models, especially in identifying disorders such as esophagitis, polyps, and ulcerative colitis—diseases where early detection is vital. Applied to SAVE-transformed pictures, EfficientNet-B2 and EfficientNetV2-B0 particularly showed the best classification accuracies. Moreover, SAVE-enhanced pictures showed better diagnostic sensitivity and specificity without calling for any more endoscopic apparatus.

## 2. Materials and Methods

### 2.1. Dataset

This work classifies and evaluates GID using the publicly accessible Kvasir v2 dataset [30]. Vestre Viken Health Trust in Norway collected the Kvasir v2 data, which consists of 6490 endoscopic images carefully categorized into six different categories: Normal (pylorus, Z-line, and cecum), esophagitis, polyps, ulcerative colitis, dyed-lifted polyps, and dyed-resection margins. These categories were selected for their clinical significance to help to diagnose and identify early GID. From the identification of subtle mucosal anomalies in normal anatomy to the identification of inflammation and precancerous lesions, every class offers distinct diagnostic problems. With more normal and polyp pictures than ulcerative colitis and dyed-resection margins, the distribution of the dataset was first quite skewed between classes. A balanced training strategy was used to improve the generalizability of trained models and lessen the effect of class imbalance. Arbitrarily split into training, validation, and test sets in line with a 7:2:1 ratio, the dataset guaranteed that every class was fairly represented over all data splits, thereby lowering the possible sampling bias during model building and evaluation. Ensuring fit with deep learning architectures depended on first preprocessing pictures. For models including ResNet and EfficientNet, all images were cropped to fit the usual input dimensions of 224 by 224 pixels. Standardizing the pixel intensity data to a [0, 1] range allowed one to expedite and stabilize model convergence during training. Data augmentation techniques were applied extensively to improve the resilience of the model and, hence, the variation of the training set. Among the enhancing methods included random rotations of up to ±15 degrees, shearing transformations of up to 0.2, zoom operations within a 0.2 range, horizontal flips, and tiny width and height alterations of 0.1. By simulating different lighting conditions and angles seen during actual endoscopic procedures, these changes created a variety of training data. For minority groups especially, augmentation helped to improve their representation and enable the model to acquire invariant traits necessary for accurate classification.

These preprocessing and augmentation strategies helped to optimize the dataset, enabling dependable approaches that support the training and evaluation of deep learning models. A brief workflow of the research is depicted in Figure 1. This figure illustrates the detailed process of the SAVE algorithm, from data acquisition to model evaluation. The workflow begins with the collection of raw white light imaging (WLI) data, followed by preprocessing steps such as resizing, cropping, and normalization of images to standard dimensions suitable for deep learning models. Data augmentation techniques, including rotations, shearing, zooming, and flipping, are employed to enhance dataset diversity and improve model robustness. The SAVE algorithm then applies spectral reconstruction techniques, leveraging polynomial regression and principal component analysis (PCA) for hyperspectral imaging (HSI) conversion. Narrow-band imaging (NBI) simulation is also performed to highlight vascular and mucosal features critical for GI disease diagnosis. The enhanced images undergo evaluation using deep learning architectures, including ResNet-50, ResNet-101, EfficientNet-B2, EfficientNet-B5, and EfficientNetV2-B0, to classify six key GI conditions. Performance metrics such as accuracy, precision, recall, and F1-score are calculated to compare SAVE-enhanced images with conventional WLI. The final output demonstrates the effectiveness of SAVE in improving diagnostic accuracy across various GI disease classes.

### 2.2. Spectrum Aided Vision Enhancer

Any color evident to the human eye can be represented through RGB values, with various combinations of red, green, and blue yielding different colors. However, the HSI model considers not only color values, but also the intensity of absorbed and reflected light. The SAVE technique converts RGB photos taken by a digital camera into HSI images by calculating a reflectance chart. The Macbeth Color Checker, commonly known as the X-Rite Classic, helps with this calibration process. This tool has 24 color patches, including the fundamental colors (red, green, and blue), secondary colors (cyan, magenta, and yellow), and six shades of gray. The 24-color patch images are transformed to the CIE 1931 XYZ color space, which normalizes and linearizes the RGB values to better match human color perception. Because images taken by digital cameras may contain noise or errors, a variable matrix correction, as indicated in Equation (1), is used. After rectification, the new X, Y, and Z values were determined using Equation (2):(1)$$\left[C\right]=\left[{XYZ}_{Spectrum}\right]\times pinv(\left[V\right])$$(2)$$\left[{XYZ}_{Correct}\right]=\left[C\right]\times [V]$$

The algorithm enables the conversion of color data from both the camera and the spectrometer into the XYZ color space. Specifically, sRGB values are mapped to the XYZ gamut for camera images, while the spectrometer’s reflectance spectra are similarly converted, following Equations (3)–(6):(3)$$X=k\int_{400nm}^{700nm}S\left(\lambda \right)R\left(\lambda \right)\bar{x}\left(\lambda \right)d\lambda$$(4)$$Y=k\int_{400nm}^{700nm}S\left(\lambda \right)R\left(\lambda \right)\bar{y}\left(\lambda \right)d\lambda$$(5)$$Z=k\int_{400nm}^{700nm}S\left(\lambda \right)R\left(\lambda \right)\bar{z}\left(\lambda \right)d\lambda$$(6)$$k=100/\int_{400nm}^{700nm}S\left(\lambda \right)\bar{y}\left(\lambda \right)d\lambda$$

The dark current component of the imaging device is accounted for by a fixed value. Standardizing the products of Vcolor, VNon-linear, and VDark—restricted to the third order to prevent over-correction—yields the variable matrix V. For color space conversion, the Ocean Optics QE65000 spectrometer and the X-Rite 24-color patch board are employed. Initially, the spectrometer measures the reflectance spectra of the 24-color patches to determine their XYZ values. A regression analysis is then performed to establish an optimized transformation matrix (M), reducing conversion errors and compensating for sensor-specific variations. A subsequent regression further enhances the alignment between estimated and reference XYZ values:(7)$$\left[M\right]=\left[Score\right]\times pinv(\left[V_{Color}\right])$$

A color transformation matrix is derived using the reflectance spectrum data (Rspectrum), with a similarity score evaluating the match between estimated and spectrometer-derived XYZ values. PCA applied to Rspectrum identifies six principal components (PCs), explaining 99.64% of the variance. The sRGB-to-XYZ transformation matrix is primarily based on the standard CIE 1931 values. Calibration experiments conducted under controlled lighting conditions refined this transformation matrix, accounting for the spectral and sensor-specific characteristics of the camera, thus ensuring robust color space conversion for SAVE image analysis. The transformation matrix correlated strongly with the PCA components, resulting in an RMSE of 0.056 and a color difference of 0.75, demonstrating high color fidelity [31,32]. The calibration significantly reduced the average chromatic aberration from 10.76 to 0.63, improving overall color accuracy. Analysis of the major color blocks indicated that red exhibited the largest deviation at wavelengths between 600 and 780 nm. Nonetheless, the remaining 23 color blocks showed RMSE values below 0.1, with black having the lowest RMSE of 0.015 and the average RMSE being 0.056, confirming excellent color reproduction. Visual and numerical assessments yielded a mean color difference of 0.75, underscoring the visual accuracy achieved.

Traditional WLI faces challenges in detecting GID; thus, specific wavelength bands are utilized to highlight affected regions, facilitating earlier diagnosis. This approach employs HSI conversion to transform RGB images into NBI equivalents, compatible with Olympus cameras for GID detection. Calibration using the 24-color checker ensures minimal discrepancy between SAVE-generated and real NBI images, with the CIEDE 2000 color difference evaluated at 2.79. Following color matching, three primary factors contributing to residual differences are addressed: the color matching function, the lighting function, and the reflection spectrum. A significant intensity disparity between 450 and 540 nm was observed due to higher light absorption. This spectral variation was calibrated using the Cauchy–Lorentz visiting distribution and optimized via fast simulated annealing (FSA), as defined in Equation (8):(8)$$f(x;x_{0},\gamma )\;=\;\frac{1}{\pi \gamma [1\;+\;{\left(\frac{x-x_{0}}{\gamma }\right)}^{2}]}\;=\;\frac{1}{\pi }[\frac{\gamma }{{\left(x\;-\;x_{0}\right)}^{2}\;+\;\gamma^{2}}]$$

With this adjustment, the color difference was down to 5.36, which is hardly noticeable. Brown tones at 650 nm were also founsd in genuine NBI pictures, despite the dominating haemoglobin absorption peaks at 415 nm and 540 nm. Hence, to improve skin cancer diagnosis and handle small artifacts caused by post-processing, more wavelengths at 600 nm, 700 nm, and 780 nm were added. With this change, calibrated NBI photos looked more like the real thing. With an average entropy of 0.37%, the SSIM of SAVE pictures reached 94.27% after calibration. Thanks to the 27.88 dB PSNR of Olympus pictures, the spectral conversion technique proved to be accurate and effective for medical imaging applications (Appendix A have important color tables as well as important figures related to SAVE).

### 2.3. Model Architecture

The selection of models for this experiment was driven by the need to achieve a stable balance between computational efficiency and classification performance. Based on preliminary research findings, the ResNet and EfficientNet architectures were chosen, as they demonstrated superior detection capabilities and higher accuracy in the SAVE process compared to other architectures, such as DenseNet and GoogleNet [33,34]. Their proven robustness, optimized feature extraction, and adaptability to different image complexities made them ideal candidates for achieving reliable and efficient performance in the targeted application. The focus on these architectures ensures a strong foundation for subsequent experiments and analyses.

#### 2.3.1. ResNet-101 and ResNet-50

ResNet (Residual Network) is a deep learning architecture solving the problem of vanishing gradients that occur while training intense neural networks. It uses residual blocks that add the previous layer outputs to the current output layer [35,36]. ResNet architecture allows the network to go very deep, with some variants containing hundreds of layers (e.g., ResNet-50, ResNet-101, ResNet-152). The network’s depth enables it to learn more complex characteristics. Mathematically, this can be represented by the following:Y = F(xWi) + x(9)
where x represents the residual block for the feature map; F(x, Wi) denote convolutional layers with weight parameters Wi through transformation; Y is the output of the feature map; and +x term ensures gradient flows directly, mitigating the vanishing gradient issues. ResNet-101, a deep convolutional neural network with 101 layers, provides greater accuracy to complicated algorithms, while ResNet-50, with 50 layers, is more efficient. ResNet-101 offers greater accuracy in deep image recognition while being more computationally costly than ResNet-50. In deep networks, training fails because the gradients of the loss function to the weights may become minuscule. By allowing gradients to pass directly through the network during backpropagation, residual connections help to alleviate the vanishing gradient problem.

#### 2.3.2. EfficientNet-B2, EfficientNet-B5, and EfficientNetV2-B0

EfficientNet is a compound scaling approach to balance depth, width, and resolution, which Mingxing Tan and Quoc V. Le introduced for achieving high accuracy by surpassing conventional CNN and using fewer parameters and less computational costs with architectures like ResNet, VGG, and Inception [37,38,39,40]. The compound scaling equation is as follows:d = αϕ, w = βϕ, r = γϕ (10)
where the constants α, β, and γ determine a tiny grid. The user-specified coefficient φ intuitively determines how many additional resources are available for model scaling. In addition, α, β, and γ indicate how these additional resources should be allocated to network width, depth, and resolution, independently. The inverted bottleneck residual blocks of MobileNetV2 lay the base foundation of the EfficientNet-B0 network, consisting of squeeze-and-excitation blocks. EfficientNetV2-B0 is a variant of EfficientNetV2, which improves upon EfficientNet by using faster training, improved scaling, and reduced parameter redundancy, and a comparison between the EfficientNet model is stated in Table 1. It offers a good mix between efficiency and precision, which makes it a viable choice for applications requiring precise object detection.

## 3. Results

This study examines the efficacy of the SAVE algorithm using different ResNet and EfficientNet machine-learning models. A comprehensive analysis focusing on metrics like precision, recall, F1-score, and accuracy is shown in Table 2, Table 3, Table 4, Table 5, and Table 6, respectively. The most successful models were EfficientNet-B2 and EfficientNetV2-B0, which outperformed ResNet-50, ResNet-101, and EfficientNet-B5 in accuracy. An improvement in the accuracy, precision, recall, and F1-score—representing total prediction of correctness across several classes—highlighted this higher execution. In accuracy, the ResNet-101 and ResNet-50 models produced comparable results, which are shown in Figure 2.

### 3.1. ResNet-101

ResNet-101 established distinct patterns when implemented with SAVE and WLI. SAVE demonstrates an improvement in precision for esophagitis, polyps, and normal classes, indicating lower false positive cases. However, WLI outperforms SAVE in dyed-resection margins and polyps in terms of recall, reducing false negative cases. F1-score, averaging precision and recall, favors SAVE in esophagitis, and polyps with an increase of 3% and 7%, respectively, while for dyed-lifted polyps, the value equals both WLI and SAVE, showing no significant increment or decrement. In cases where higher recall is more relevant, SAVE would be more productive as it ensures a lower rate of missed cases, which can help in early detection and intervention of GID, and the results are shown in Table 2 (Appendix A shows the confusion matrix, and Appendix A shows the plots).

**Table 2 bioengineering-12-00828-t002:** WLI and SAVE evaluation on ResNet-101 architecture.

Dataset	Classes	Precision	Recall	F1-Score	Accuracy
WLI	Dyed-Lifted Polyps	85%	73%	78%	83%
	Dyed-Resection Margins	89%	77%	82%	
	Esophagitis	75%	80%	77%	
	Normal	84.11%	89.30%	86.36%	
	Polyps	74%	83%	78%	
	Ulcerative Colitis	91%	83%	87%	
SAVE	Dyed-Lifted Polyps	86%	76%	81%	85%
	Dyed-Resection Margins	78%	78%	78%	
	Esophagitis	85%	75%	80%	
	Normal	87.81%	89.67%	88.90%	
	Polyps	84%	86%	85%	
	Ulcerative Colitis	77%	89%	82%	

### 3.2. EfficientNet-B2

Evaluating the performance of the EfficientNet-B2 architecture on SAVE images, it exceeded the performance of WLI across dyed-lifted polyps by 3% and 11% in recall and precision, suggesting that SAVE is more effective in reducing false positives and false negatives with an increase in true positives. However, WLI performed better in dyed-resection margins, with a precision of 90% compared to 83% in SAVE, demonstrating an equal F1-score of 83%, showing that both have the same impact, and the results are shown in Table 3. In clinical settings where true positives play a major role in diagnosis, WLI is preferable, helping in the reduction in false positives during predictions (Appendix A shows the confusion matrix, and Appendix A shows the plots).

**Table 3 bioengineering-12-00828-t003:** WLI and SAVE evaluation on EfficientNet-B2 architecture.

Dataset	Classes	Precision	Recall	F1-Score	Accuracy
WLI	Dyed-Lifted Polyps	77%	83%	79%	85%
	Dyed-Resection Margins	90%	77%	83%	
	Esophagitis	81%	76%	78%	
	Normal	86.19%	89.14%	89.37%	
	Polyps	79%	85%	82%	
	Ulcerative Colitis	96%	78%	86%	
SAVE	Dyed-Lifted Polyps	86%	80%	83%	86%
	Dyed-Resection Margins	83%	82%	83%	
	Esophagitis	80%	81%	81%	
	Normal	87.73%	89.81%	89.53%	
	Polyps	84%	86%	85%	
	Ulcerative Colitis	93%	84%	88%	

### 3.3. ResNet-50

With ResNet-50 architecture, SAVE performed better than WLI in terms of precision, recall, and F1-score. SAVE performs significantly better in the precision of polyps, achieving an increase of 14%, but this contrasts with the decrease of 12% in dyed-resection margins. In the case of ulcerative colitis, WLI outperforms by achieving 98% precision compared to 91% in SAVE, but a better balance in recall by an increment of 4% is seen in SAVE; however, the F1-score remains the same for both balancing precision and recall, and the results are shown in Table 4. From an application perspective, SAVE would be preferable in conditions where high recall is required, where false negatives can lead to higher risks of false diagnosis of a true positive (Appendix A shows the confusion matrix of ResNet-50, and Appendix A shows the plots).

**Table 4 bioengineering-12-00828-t004:** WLI and SAVE evaluation on ResNet-50 architecture.

Dataset	Classes	Precision	Recall	F1-Score	Accuracy
WLI	Dyed-Lifted	77%	81%	79%	83%
	Polyps	79%	85%	82%	
	Dyed-Resection Margins	89%	71%	79%	
	Esophagitis	77%	75%	76%	
	Normal	84.9%	90.5%	87.7%	
	Polyps	76%	87%	81%	
	Ulcerative Colitis	98%	81%	88%	
SAVE	Dyed-Lifted Polyps	81%	81%	81%	85%
	Dyed-Resection Margins	77%	79%	78%	
	Esophagitis	80%	76%	78%	
	Normal	87.3%	90.9%	88.8%	
	Polyps	90%	85%	87%	
	Ulcerative Colitis	91%	85%	88%	

### 3.4. EfficientNet-B5

The comparatively low accuracy, recall, and F1-score of the EfficientNet-B5 model in contrast to others point to potential architectural or training data flaws affecting its functionality. The model misclassifies negative cases as positives at a high rate while finding it difficult to identify actual positive cases, as seen by the sharp 20% decrease in the precision of class dyed-resection margins in SAVE, and the results are shown in Table 5. WLI also performs better with an increase of 10% precision in ulcerative colitis, making it preferable in scenarios that require detection of true positive cases with a lower rate of false positives (Appendix A shows the confusion matrix, and Appendix A shows the plots).

**Table 5 bioengineering-12-00828-t005:** WLI and SAVE evaluation on EfficientNet-B5 architecture.

Dataset	Classes	Precision	Recall	F1-Score	Accuracy
WLI	Dyed-Lifted Polyps	69%	85%	76%	81%
	Dyed-Resection Margins	96%	62%	75%	
	Esophagitis	70%	73%	72%	
	Normal	82.9%	88.3%	85.4%	
	Polyps	78%	86%	82%	
	Ulcerative Colitis	98%	76%	86%	
SAVE	Dyed-Lifted Polyps	77%	80%	78%	83%
	Dyed-Resection Margins	76%	72%	74%	
	Esophagitis	78%	76%	77%	
	Normal	86.5%	90.0%	88.3%	
	Polyps	83%	79%	81%	
	Ulcerative Colitis	88%	84%	86%	

### 3.5. EfficientNetV2-B0

SAVE emerges as highly compatible with EfficientNetV2-B0 as it performs impressively in increments of both precision and recall values. SAVE outperforms WLI in dyed-lifted polyps and polyps by increasing by 6% and 7% in precision, suggesting that it reduces false positive cases while maintaining strong sensitivity, which can help in identifying misdiagnoses. Respectively, WLI has shown a progressive increase of 98% in the precision of ulcerative colitis compared to 88% in SAVE. However, SAVE balances the recall with 8%, achieving an F1-score of 89%, which equals that of WLI. In endoscopy, classification plays a vital role in identifying different GIDs, and SAVE implemented with EfficientNETV2-B0 serves better results than WLI, and the results are shown in Table 6, thereby serving as a better option for endoscopic classification (Appendix A shows the confusion matrix, and Appendix A shows the plots).

**Table 6 bioengineering-12-00828-t006:** WLI and SAVE evaluation on EfficientNetV2-B0 architecture.

Dataset	Classes	Precision	Recall	F1-Score	Accuracy
WLI	Dyed-Lifted Polyps	76%	85%	80%	84%
	Dyed-Resection Margins	93%	71%	81%	
	Esophagitis	76%	75%	76%	
	Normal	83.5%	92.2%	89.4%	
	Polyps	77%	85%	81%	
	Ulcerative Colitis	98%	82%	89%	
SAVE	Dyed-Lifted Polyps	82%	81%	82%	86%
	Dyed-Resection Margins	85%	82%	83%	
	Esophagitis	81%	81%	81%	
	Normal	89.8%	91.2%	90.5%	
	Polyps	84%	84%	84%	
	Ulcerative Colitis	88%	90%	89%	

Consequently, it is anticipated that the SAVE software will be easily transferable as a real-time plugin in routine endoscopic practice, enabling the conversion of conventional WLI into HSI and NBI-augmented views in real time to identify subtle mucosal and vascular abnormalities as they arise during endoscopy. SAVE simulated NBI contrast may improve the early detection of dysplastic alterations without requiring specialized imaging equipment in the monitoring of high-risk populations, such as patients with Barrett’s esophagus or chronic inflammatory bowel disease. The offline analysis of capsule endoscopy video is expected to enhance the visibility of lesions and reduce reading times by emphasizing the vascular patterns typical of angioectasias or neoplastic foci. Alongside human interpretation, processed SAVE frames can be integrated into CAD algorithms for polyp detection, inflammation scoring, or dysplasia grading, as the spectral features may enhance sensitivity and specificity. As a solely software-based system, SAVE democratizes access to advanced contrast modes that are typically inaccessible in resource-constrained settings lacking specialized endoscopes, and can facilitate guided procedures, potentially leading to diminished recurrence rates. The incorporation of SAVE into educational and tele-endoscopy programs utilizing cloud computing will standardize training, providing access to experts irrespective of geographical constraints, ultimately enhancing the quality and consistency of gastrointestinal disease management across diverse clinical environments. To facilitate the implementation of SAVE in routine endoscopy, the framework is evaluated on a clinical-grade workstation, achieving an average per-frame latency from 20 ms to 30 ms, which is significantly below the requirements for live video performance. A single-center pilot study will be conducted that will involve the simultaneous operation of SAVE alongside the standard WLI during diagnostic procedures. The primary endpoints will include the rate of lesion identification, the time required for lesion delineation, the biopsy yield, and the impact on the endoscopists’ workflow, which will be gathered using structured case report forms. A multi-center prospective trial is being designed to compare the diagnostic accuracy, procedure duration, and user satisfaction of SAVE augmented workflow versus conventional WLI workflow, based on pilot data. To facilitate medical interpretation, a concise visual template and training module are created to feature images of typographical SAVE in intensity and vascular patterns. Collaboration with endoscopy nurses, pathologists, and regulatory specialists will ensure rigorous testing of technical performance and clinical usability, facilitating the pathway to regulatory submissions and the extensive utilization of limited resources.

## 4. Discussion

This work presents and tests the SAVE, a computational system intended to supplement GI endoscopic imaging by reconstructing HSI and simulated NBI data from traditional WLI. This method addresses the long-standing constraints of conventional endoscopic modalities in identifying subtle mucosal and vascular anomalies vital for early-stage GID diagnosis by using a combination of spectral calibration, PCA, and advanced regression approaches [41]. The results of this study highlight the SAVE capacity to greatly improve diagnostic performance without resorting to costly or specialized hardware upgrades—a matter of great relevance for clinical scalability and access. Despite the majority of conditions demonstrating consistent enhancements following SAVE implementation, the dyed-resection margins category shows a decline in precision of 96% (WLI) and 76% (SAVE) when utilizing EfficientNet-B5. This is presumably attributable to two factors. The spectral reconstruction utilizing polynomial regression may alter the specific hue and saturation of colors to which EfficientNet-B5 was trained to react favorably, despite a minor shift in spectral measurements, potentially leading to inadvertent false positives not representative of dye uptake. Secondly, the NBI imaging effect simulated through the Lorentzian filtering method could accentuate the vascular structures beyond the dye-specific contrast, thereby complicating the margin delineation. As a solution, these SAVE versions may incorporate a class-adaptive calibration step, which either adjusts the polynomial coefficients on margin-specific patches or implements a straightforward re-calibration network that substitutes the original WLI color cues of this class. Over several deep learning architectures—including ResNet-50, ResNet-101, EfficientNet-B2, EfficientNet-B5, and EfficientNetV2-B0—the use of SAVE considerably enhanced classification accuracy, precision, recall, and F1-scores [42]. This consistent performance improvement across several CNN models implies that the spectral and spatial information generated by the SAVE reconstruction techniques offers richer feature sets that are easily accessible by current deep learning classifiers. Especially EfficientNet-B2 and EfficientNetV2-B0 produced the best accuracies, which matched the architectural efficiencies and improved the feature extraction power inherent in these models. These findings align with earlier studies showing that compound scaling in EfficientNet variants maximizes depth and resolution, thereby enabling better performance on challenging biomedical imaging applications without disproportionate processing overhead.

Several technical developments included in the algorithm help to explain the better performance of SAVE-enhanced photos. First, the color calibration process using the Macbeth Color Checker and XYZ color space transformations greatly reduced chromatic aberrations; therefore, preserving color integrity is vital for replicating diagnostically salient features including vascular patterns and mucosal texture. Second, the use of PCA and polynomial regression allowed for very accurate spectrum reconstruction, achieving an RMSE of 0.056 and an SSIM above 90%. These measures not only show the accuracy of the algorithm, but also its capacity to preserve the structural integrity of the original images, which is essential for clinical relevance in which visual artifacts might undermine diagnosis validity. Furthermore, the SAVE ability to artificially create narrow-band representations faithfully replicates the contrast enhancement usually attained by dedicated NBI systems, which emphasize vascular structures by means of haemoglobin absorption peaks [43]. Particularly in identifying early-stage oesophagitis, polyps, and ulcerative colitis—conditions where early intervention can significantly change disease trajectories—the results of the present investigation show that SAVE-generated NBI images enhanced both precision and recall. Significantly, this was accomplished without requiring proprietary NBI-compatible endoscopes, hence democratizing access to sophisticated imaging modalities in limited resources.

Although SAVE showed strong performance throughout most classes, it is interesting that performance variance persisted in some areas, like colored resection margins, especially in relation to EfficientNet-B5. When used on datasets with imbalanced class distribution or modest inter-class variations, this variability most likely reflects the architectural restrictions of the EfficientNet-B5 model. This emphasizes the need for choosing architecture fit for clinical datasets and the possibility for even more optimization by means of class-balanced loss functions or transfer learning. Furthermore, the current work’s use of the publicly accessible Kvasir v2 dataset imposes certain restrictions, despite its benefits for reproducibility and benchmarking. Applied to more heterogeneous real-world clinical datasets, the natural class imbalance of the dataset may affect model generalizability even with data augmentation and stratified sampling mitigating measures. Larger, multicentre datasets and prospective validation studies should be included in future studies to evaluate the SAVE clinical value over several patient demographics and endoscopic systems.

Moreover, even if the SAVE framework now runs offline, integration into real-time clinical processes is vital. Real-time implementation would need the optimization of computational efficiency, maybe by means of model quantization or deployment on specialized hardware accelerators. These developments would allow for dynamic upgrading of endoscopic feeds and provide doctors with instant diagnostic support during operations. Finally, by enhancing WLI with hyperspectral and NBI-like information, this work offers strong proof that the SAVE method greatly increases the diagnostic capacity of GI endoscopy. SAVE has shown promise as a transforming tool in computational endoscopy by means of thorough quantitative validation and comparison assessments with state-of-the-art CNN designs. To completely realize the SAVE promise in improving early detection and clinical outcomes in GID treatment, future research initiatives should concentrate on real-time deployment, external validation, and expansion to other imaging modalities and anatomical locations.

## 5. Conclusions

This study introduces the SAVE, a new and scalable method for improving endoscopic imaging of the GI tract. SAVE effectively converts conventional WLI into HSI and NBI representations by combining spectral reconstruction using polynomial regression and PCA with targeted narrow-band simulation using Lorentzian filtering. While maintaining important diagnostic information like mucosal and vascular features, the method showed outstanding reconstruction ability and achieved low RMSE values and high SSIM scores. With their strong resemblance to the optical behavior of specialist NBI systems, the SAVE simulated NBI significantly increases lesion visibility without requiring specific tools. Comparative studies confirmed that SAVE is quite flexible for both conventional and capsule endoscopy uses since it preserves excellent structural integrity and signal quality. Crucially, the SAVE software-driven approach democratizes access to sophisticated imaging modalities, thereby benefiting a wider spectrum of healthcare situations. Future projects will concentrate on real-time integration into clinical processes and outside validation across several patient groups. With great potential to raise early detection, increase diagnostic accuracy, and finally improve clinical outcomes in gastrointestinal disease care, SAVE represents a major advancement in computational endoscopy.

## Figures and Tables

**Figure 1 bioengineering-12-00828-f001:**
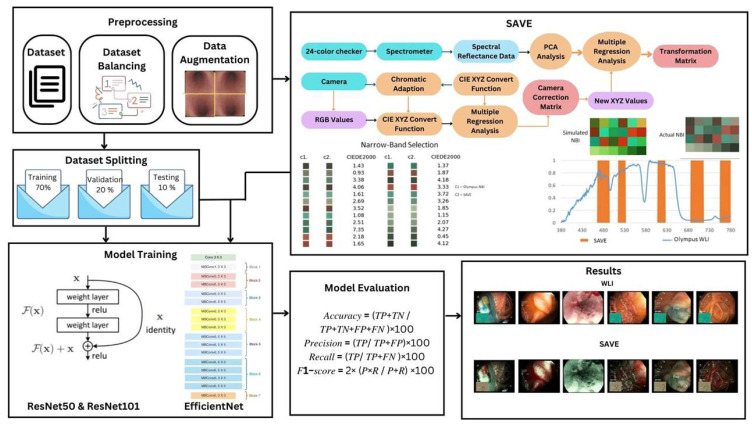
Workflow of the SAVE framework for enhancing gastrointestinal (GI) disease detection.

**Figure 2 bioengineering-12-00828-f002:**
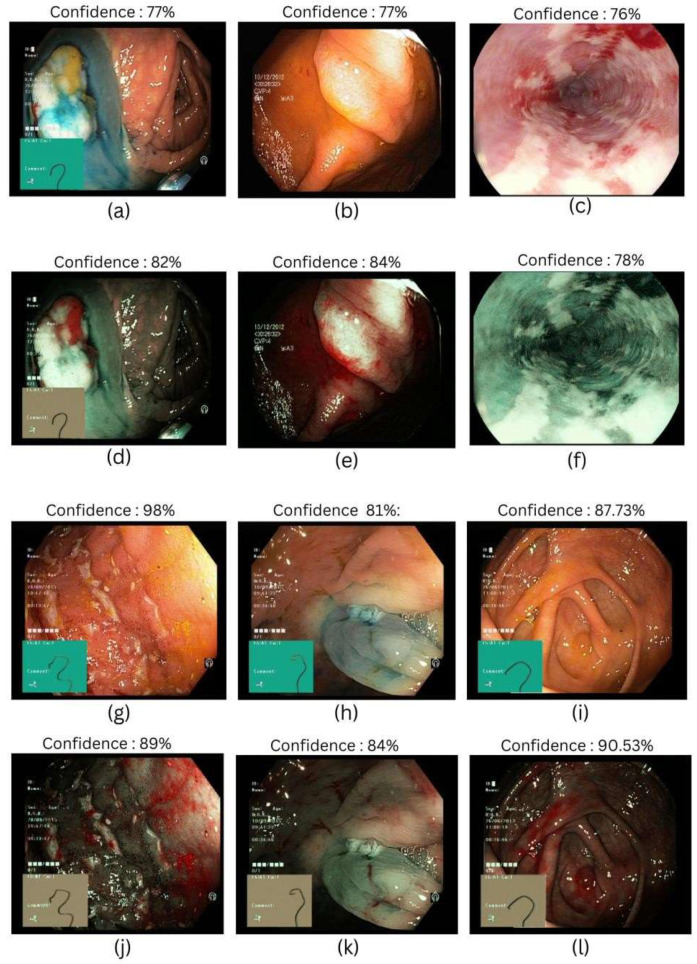
Different classes of gastrointestinal diseases were used in this study. (**a**,**d**) Represent dyed-lifted-polyps in WLI and SAVE, respectively; (**b**,**e**) represent polyps in WLI and SAVE, respectively; (**c**,**f**) represent esophagitis in WLI and SAVE, respectively; (**g**,**j**) represent ulcerative colitis in WLI and SAVE, respectively; (**h**,**k**) represent dyed-resection margins in WLI and SAVE, respectively; and (**i**,**l**) represent normal classes in WLI and SAVE, respectively.

**Table 1 bioengineering-12-00828-t001:** Comparison between EfficientNet-B2, EfficientNet-B5, and EfficientNetV2-B0.

Model	Depth (D)	Width (W)	Resolution (R)	Parameters	FLOPs (Billion)
EfficientNet-B2	1.2×	1.4×	260 × 260	9.2 M	1.0
EfficientNet-B5	2.2×	2.6×	456 × 456	30 M	9.9
EfficientNetV2-B0	16	1.2×	192 × 192	7.1 M	0.8 GFLOPs

## Data Availability

The data presented in this study are available upon considerable request to the corresponding author (H.-C.W.).

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
