# Peer review of "Enhancing Early GI Disease Detection with Spectral Visualization and Deep Learning"

_bioengineering, 2025, doi:10.3390/bioengineering12080828_

Round 1

Reviewer 1 Report

Comments and Suggestions for Authors

Dear Authors,

Thank you for allowing me to review this interesting article. Its main objective is to analyze the impact of a specific software, SAVE 27, on early detection of mucosal/ vascular changes identified by clinical endoscopy. The benefits would be represented by improved diagnosis in gastro-intestinal diseases (GIDs) when compared to that obtained using conventional white light imaging (WLI).  Advanced spectral reconstruction techniques are introduced by SAVE. Six main GID conditions have been investigated using this novel technique. 

The introduction provides main issues related to the proper detection of GDIs using clinical endoscopy. Fig. 1 gives a proper description of the place SAVE can have in the diagnosis of GIDs. The methodology of the article is well described. The results demonstrate that SAVE constantly outperformed WLI across several critical issues, without any need for specialized imaging hardware.  The statistical analysis section is well documented and supports the results. The Discussion section is also appropriate. There is enough data to support the conclusions, which basically support the utility of SAVE in getting early detection and broader access to advanced imaging, especially in resource-constrained settings.

I would like to see some suggestions related to the clinical use you anticipate for SAVE. 

There are no other suggestions I would like to make, since the article is technically sound and well supported by data.  

Reviewer 2 Report

Comments and Suggestions for Authors

The manuscript presents and assesses Spectrum Aided Vision Enhancer (SAVE), which is an innovative, software-driven framework that transforms standard white light imaging (WLI) into high-fidelity hyperspectral imaging (HSI) and simulated narrow-band imaging (NBI) without any hardware modification. 
Results demonstrated that SAVE enhanced imagery consistently outperformed raw WLI across precision, recall, and F1-score metrics, with DL models such as EfficientNetB2 and EfficientNetV2-B0 achieving the highest classification accuracy.
SAVE could become a transformative solution for augmenting gastrointestinal diseases diagnostics, with the potential to significantly improve early detection, streamline clinical workflows, and broaden access to advanced imaging especially in resource constrained settings.
I find the topic interesting and being worth of investigation.
The manuscript is fluidly written and has a strong organization. The introduction provides enough background, the methodology approach is detailed and well explained.The discussion links well the research with previous knowledge and the conclusion is supported by the results.
Although, I carefull advise authors in the following matters:
- Order the keywords alphabetically.
- At abstract "...presents SAVE (Spectrum Aided Vision Enhancer)..." should be chaged to "...presents Spectrum Aided Vision Enhancer (SAVE)..."
- Since the manuscript uses some acronyms, probably it is a good idea to have an glossary
- The Kvasir v2 dataset (lines 119-121) should be described in the second section (Materials and Methods) and not in the introduction 
- At the end of intruduction section the aim of the research should be outlined
- Tables should not be broken in 2 pages
- Results should be presented using the ROC analysis and have the AUC chart, in order to enhance the diagnostic power
- I strongly suggest authors from refraining using personal pronouns such as "we" and "our" throughout the text and I encourage them to write it in an impersonal form of writing.

Comments on the Quality of English Language

I strongly suggest authors from refraining using personal pronouns such as "we" and "our" throughout the text and I encourage them to write it in an impersonal form of writing.

Reviewer 3 Report

Comments and Suggestions for Authors

The article proposes a novel computational solution (SAVE) that simulates hyperspectral (HSI) and narrow-band (NBI) images from standard endoscopic white light images (WLI), without requiring any hardware modifications. The tool improves sensitivity in the early diagnosis of gastrointestinal diseases (GID), which is of great clinical interest.

In my opinion, the following aspects should be improved:

Dataset limitation: Although the use of Kvasir v2 is appropriate, the generalization to real clinical settings is not addressed in sufficient depth (it is briefly mentioned at the end, but not developed).

Lack of prospective clinical validation: Although SAVE improves image performance, there is no discussion on how it could be integrated into actual clinical protocols (workflow, processing time, medical interpretation).

Comparison with real NBI: SAVE is not compared with NBI images obtained from actual equipment, which would help justify its clinical use as a substitute.

Precision limitations in certain models: In EfficientNet-B5, SAVE performs worse in some classes (e.g., dyed resection), which deserves more in-depth discussion.

The level of English is technically correct, although at times the tone is redundant or unnecessarily emphatic (e.g., “SAVE presents a fresh and useful way to augment…”).

There are stylistic and punctuation issues that should be corrected, such as unnecessary repetitions (e.g., “SAVE novelty” is used excessively), and inconsistent use of verb tenses (present/past). Caution is also advised, as abbreviations like WLI, HSI, and NBI are not defined at their first mention in the abstract (although they are in the main text).

To improve the manuscript, clinical limitations should be explored in greater depth: explicitly addressing how SAVE could be integrated into real-world clinical workflows. A direct comparison with actual NBI images (if available) would also be of interest, as it would strengthen the clinical validity of the approach. The fluency of some sections could be improved—particularly the "Materials and Methods" section, which is technical but somewhat dense and fragmented.
